# Effectiveness of Se/ZnO NPs in Enhancing the Antibacterial Activity of Resin-Based Dental Composites

**DOI:** 10.3390/ma15217827

**Published:** 2022-11-06

**Authors:** Iqra Saleem, Nosheen Fatima Rana, Tahreem Tanweer, Wafa Arif, Iqra Shafique, Amenah S. Alotaibi, Hanadi A. Almukhlifi, Sohad Abdulkaleg Alshareef, Farid Menaa

**Affiliations:** 1Department of Biomedical Engineering and Sciences, School of Mechanical and Manufacturing Engineering, National University of Sciences and Technology, Islamabad 44000, Pakistan; 2Genomic and Biotechnology Unit, Department of Biology, Faculty of Science, University of Tabuk, Tabuk 71491, Saudi Arabia; 3Department of Chemistry, Faculty of Science, University of Tabuk, Tabuk 71491, Saudi Arabia; 4Department of Biology, Faculty of Science, University of Tabuk, Tabuk 71491, Saudi Arabia; 5Departments of Internal Medicine and Nanomedicine, California Innovations Corporation, San Diego, CA 92037, USA

**Keywords:** antibacterial, secondary caries, restoration, microcosm, composite resin

## Abstract

Biofilm formation in the resin-composite interface is a major challenge for resin-based dental composites. Using doped z nanoparticles (NPs) to enhance the antibacterial properties of resin composites can be an effective approach to prevent this. The present study focused on the effectiveness of Selenium-doped ZnO (Se/ZnO) NPs as an antibacterial nanofiller in resin composites and their impact on their mechanical properties. Pristine and Se/ZnO NPs were synthesized by the mechanochemical method and confirmed through UV-Vis Spectroscopy, FTIR (Fourier Transform Infrared) analysis, X-ray Diffraction (XRD) crystallography, Scanning Electron Microscopy (SEM), Energy Dispersive Spectroscopy (EDS), and Zeta analysis. The resin composites were then modified by varying concentrations of pristine and Se/ZnO NPs. A single species (*S. mutans and E. faecalis*) and a saliva microcosm model were utilized for antibacterial analysis. Hemolytic assay and compressive strength tests were also performed to test the modified composite resin’s cytotoxicity and mechanical strength. When incorporated into composite resin, 1% Se/ZnO NPs showed higher antibacterial activity, biocompatibility, and higher mechanical strength when compared to composites with 1% ZnO NPs. The Se/ZnO NPs has been explored for the first time as an efficient antibacterial nanofiller for resin composites and showed effectiveness at lower concentrations, and hence can be an effective candidate in preventing secondary caries by limiting biofilm formation.

## 1. Introduction

Secondary caries is a common oral infection similar to primary caries and usually appears on the tooth’s gingival side [1]. Microcrack formation between the restoration–tooth interface is the leading cause of bacterial microleakage and resultant biofilm formation [2]. Secondary caries, if not treated, eventually lead to the replacement of restorative material, hence adding to health care expenses [1]. In the United States (US), the estimated annual cost of restorative materials in 2005 was USD 46.2 billion, which was expected to reach USD 49.7 billion [3,4]. Globally, direct dental restorations treatment costs are projected to be USD 298 billion yearly, which is around 4.6% of total health expenditures [5]. Hence, the focus was shifted towards the development of restorations that are long-lasting in order to avoid associated costs.

Recently, resin-based composites have gained advancement and popularity due their physical and mechanochemical properties [6,7], including their aesthetics and improved mechanical and chemical binding with enamel and dentin [8]. They were first used in the 1950s, and their demand increased significantly due to their superior properties [9]. Over 95% of all direct restorations on anterior teeth and 50% of all direct restorations on posterior teeth currently employ resin composites [10]. Inorganic filler and organic resin-based matrix phases comprise most of the composites. In addition to their primary use as direct restorative materials, resin-based composites are also used in other dental applications such as cement and indirect restorative materials. Resin-based composites are also used in CAD/CAM systems for machining or as a direct-placement restorative material in the form of a paste that later needs curing [9]. However, gap formation is one of the major concerns faced by composite-based restorative materials, allowing the cariogenic biofilm to enter the dentine–composite resin interface [11,12,13]. Bacteria responsible for biofilm formation produce acid on the surface of the tooth through carbohydrate fermentation, demineralizing the calcified tissues of the tooth and ruining the tooth structure, and leading to secondary caries [12,13]. This increases the demand for antibacterial composites that can completely inhibit or lower the biofilm accumulation rate, thereby preventing secondary caries [12,14]. 

Multiple strategies have been proposed in the last few decades to overcome this problem [2]. One strategy was to incorporate antibiotics in dental restorative materials to reduce the adherence of biofilm-forming pathogens on the surface of dental materials [14]. These antibiotics limit biofilm accumulation, hence controlling the acid-enforced demineralization, which eventually prolongs the composite resin’s life [15,16] and limits the occurrence of secondary caries. 

Nanotechnology has played a vital role in enhancing dental restorative materials [17,18,19,20]. Antibacterial NPs such as silver (AgNPs) and metal oxides NPs have been extensively used to improve the antibacterial properties of resin-based composites [17,18,19,20]. Antibacterial nano-assemblies have also been reported to enhance the antibacterial properties of resin-based composites [17]. However, incorporating these NPs may lead to reduced biocompatibility and mechanical strength as well as aesthetics [17,18,19,20]. Among metal oxide NPs, Zinc Oxide (ZnO) NPs have been reported to be as effective antibacterial metal oxide NP [12,20]. It has been deemed an antibacterial nanofiller for dental composites to prevent secondary caries formation [12,20]. Incorporating ZnO NPs in commercially available composite resin enhances antibacterial properties by releasing Zinc ions (Zn^2+^) in the tooth-composite interface, thus inhibiting biofilm accumulation and preventing secondary caries [21]. However, it is reported that ZnO NPs have limited antibacterial activity at lower concentrations, whereas increasing the concentration deteriorates the mechanical properties of the composites [14,22]. To enhance the antibacterial activity of ZnO, doping it with metals has been reported as an effective approach [23]. 

Selenium (Se) is a metalloid with high antibacterial potential and when doped onto ZnO, it has been reported to enhance the antibacterial potential of the ZnO NPs [24]. However, Se-doped ZnO NPs have not been explored in dental applications. Their effectiveness as antibacterial activity enhancers of dental composites has not been previously explored. This study aimed to explore for the first time the effectiveness of Se-doped Zinc Oxide NPs (Se/ZnO NPs) as an antibacterial activity enhancer of composite resin against oral microorganisms.

## 2. Materials and Methods

All the chemicals used in this research were purchased from Sigma Aldrich unless indicated otherwise.

### 2.1. Synthesis of Pristine ZnO NPs and Se/ZnO NPs

Pristine and Se/ZnO NPs were synthesized by the mechanochemical reaction. Pristine ZnO NPs were first synthesized by manually mixing 25 mmol or 5.48 g of zinc acetate dihydrate (Zn (CH_3_COO)_2_·2H_2_O) and 42 mmol 3.78 g of oxalic acid (C_2_H_2_O_4_·2H_2_O) with a pestle and mortar at room temperature. The mixing and grinding continued until the smell of acetic acid (CH_3_COOH) stopped, indicating the reaction’s completion. After 30 min, hydrated zinc oxalate (ZnC_2_O_4_) was formed. The precursors were washed with ethanol multiple times, and each step involved manual shaking and 10 min of sonication. Afterward, the NPs were oven dried at 40 °C for 12 h. Dried NPs were annealed at 450 °C for 30 min in a temperature-controlled furnace to obtain fine crystalline pristine ZnO NPs [25]

Similarly, Se/ZnO NPs were obtained by adding 0.5 mmol of Se powder (0.04 g) to the mixture of 5.48 g of zinc acetate dihydrate (Zn (CH_3_COO)_2_ 2H_2_O) and 3.78 g of oxalic acid (C_2_H_2_O_4_·2H_2_O) using a pestle and mortar and grinding at room temperature. The same procedure was followed to obtain Se/ZnO NPs [25]. 

### 2.2. Physical Characterization

The synthesized pristine and Se/ZnO NPs were confirmed by several characterization techniques routinely used [15,25,26,27,28]. The first confirmation was done by UV-Vis spectrophotometry from UV-2800 BMS Biotechnology Medical Services, Madrid, Spain, spectrophotometer. FTIR was performed using Bruker FTIR Spectrometer ALPHA (Westborough, MA, USA). XRD crystallography using Benchtop X-ray Powder Diffraction (Bruker, MA, USA), SEM, and EDX were performed with SEM VEG 3 LMU (Tescan, Czech Republic) to confirm the crystalline nature. The SEM images were taken at the resolution of X50,000; for the size interpretation, the scale was set at 0.5 µm, and the Zeta potential (ZP) was analyzed by Malvern Zeta sizer (Malvern, Worcestershire, UK). 

### 2.3. Bacterial Strain Isolation

The two main clinical bacterial isolates of *S. mutans* and *E. faecalis* were utilized to perform the study. To isolate these strains from the fresh salivary sample, saliva was collected in sterilized containers from 10 volunteers after taking their consent. These volunteers neither brushed their teeth nor ate anything for at least 10 h. They also did not have any dental treatments before. These salivary samples were mixed, and different dilutions were prepared by mixing this sample in sterilized distilled water. These dilutions were spread on Tryptic Soy Agar (TSA) plates and incubated at 37 ° C for 24 h. Different colored colonies grew on these incubated TSA plates. The colonies were picked based on their color and morphological differences and streaked on TSA plates. After 24 h, colonies grown on TSA plates were picked and further streaking was done on selective media (Blood Agar plates) for our desired bacterial strain [29,30].

### 2.4. Preparation of Composite Resin Discs

Commercial composite resin, i.e., Nexcomp-META BIOMED, was used in this study. This composite resin comprised Bis-GMA, UDMA, Bis-EMA, and organic polymers. Prepared NPs in different percentages (1%, 2.5%, 5%) per weight (0.1 g) of the composite discs were manually mixed in the composite resin. Composite discs were prepared by using plastic molds (2 mm h × 4 mm d). After the preparation of the discs, these discs were cured and placed under the blue UV light having an intensity of 400 mW/cm^2^ and a wavelength of 430–480 nm for almost 4 h. Composite resin discs were prepared to measure antibacterial activity, as well as hemolysis and mechanical testing of prepared bioactive composite resin.

### 2.5. In Vitro Antibacterial Testing

#### 2.5.1. Single Species Model

*S. mutans* and *E. faecalis* broth cultures were started by inoculating 5 mL of TSB + S (1% sucrose) with colonies from the blood agar plates and incubated for 24 hrs in a 5% CO_2_ incubator. From the overnight *S. mutans* and *E. faecalis* TSBS preculture, a culture was started by inoculating 5 mL of TSBS broth with 200 µL of preculture. After 3 (*S. mutans*) and 3.5 h (*E. faecalis*), respectively, when the optical density (OD) of the culture reached 1, a 600 µL culture was diluted through serial dilutions. Then, 500 µL of 10^−5^ diluted cultures of *S. mutans* and *E. faecalis* were added to sterile centrifuge tubes. The bacteria were incubated with the composite specimens for 6 h in a 5% CO_2_ incubator. After 6 h, the centrifuge tubes were removed from the incubator and vortexed. A total of 50 µL sample from each centrifuge tube was taken, spread on TSA + S plates, and incubated for 18–24 h [26].

#### 2.5.2. Microcosm Model

Saliva samples from 10 volunteers were collected. It was then diluted with sterile glycerol to the concentration of 30% and stored at −80 °C. In total, 50 µL of saliva inoculum was added to 10 mL fresh TSBS (1% sucrose) and incubated overnight. Then, 200 µL of preculture was added to 5 mL of fresh TSBS, and the OD obtained 1 at 600 nm; it was then serially diluted until reaching 10^−5^. Afterward, 500 µL of it was added to each centrifuge tube with composite discs present in them and placed in an incubator for 6 h. After 6 h, the centrifuge tubes were removed from the incubator and vortexed. A total of 50 µL sample from each centrifuge tube was taken and spread on TSAS (1% sucrose) plates and incubated for 18–24 h [26].

#### 2.5.3. Colony Forming Units (CFU)

Colonies obtained on plates were counted manually, and CFU/mL was calculated using the formula: (1)$$\frac{\mathrm{No}.\;\mathrm{of}\;\mathrm{colonies}\;\times \mathrm{Dillution}\;\mathrm{factor}\;}{\mathrm{Sample}\;\mathrm{poured}\;\mathrm{in}\;\mathrm{each}\;\mathrm{plate}\;\mathrm{in}\;\mathrm{mL}}$$

The dilution factor used was 10^−5,^ and the sample spread in each plate was 0.05 mL. 

### 2.6. Mechanical Testing 

For testing the mechanical strength of prepared composite discs, they were placed in saliva for 24 h after curing with UV light. A universal testing machine (Instron universal testing machine model static) at a crosshead speed of 0.5 cm/min and load cell of load cell 5 KN were used to measure the compressive strength. Specimens were placed horizontally on the machine’s base, and the compressive load was applied until fracture [31]. The unit of measured strength was MPa, and the Compressive strength (CS) was calculated by using this formula:(2)$$\mathrm{CS}=\frac{\mathrm{Fracture}\;\mathrm{Load}}{\mathrm{Area}\left({\mathrm{mm}}^{2}\right)}$$

### 2.7. Biocompatability Assay

To evaluate the biocompatibility of synthesized NPs, a human blood sample was collected from a female volunteer after receiving informed consent. It was diluted with PBS in equal amounts and centrifuged for 10 min at 13,000 rpm to isolate the RBCs. This procedure was repeated twice. The isolated RBCs were diluted with PBS at a 1:2 concentration. In total, 10 mL PBS in each test tube containing 200 µL of diluted blood sample and composite discs was incubated for 2 h. Triton X (10 mL) diluted with blood (200 µL) was taken as the Positive control. The PBS solution (10 mL) with blood sample (200 µL) was taken as the negative control [30]. Two approaches were used to analyze the hemolytic activity: qualitative and quantitative. After incubation, all samples were centrifuged for 5 min at 5000 rpm [32]
(3)$$\%\;\mathrm{Hemolysis}=\frac{\left(O.D.\;\mathrm{test}\;\mathrm{sample}\right)-\left(\mathrm{OD}\;\mathrm{negative}\;\mathrm{control}\right)}{\left(\mathrm{OD}\;\mathrm{posiitive}\;\mathrm{control}\right)-\left(\mathrm{OD}\;\mathrm{negative}\;\mathrm{control}\right)}\times 100$$

### 2.8. Statistical Analysis

All the results were statistically analyzed using Graph Pad Prism software 9.2. One- way ANOVA and t test were performed for group comparisons at *p* value less than 0.05. 

## 3. Results

ZnO NPs and Se/ZnO NPs were successfully obtained in white powder form by mechanochemical method or solid-state reaction and confirmed by different characterization techniques (Figure 1a).

### 3.1. UV-Vis Analysis of Pristine ZnO NPs and Se/ZnO NPs

In this study, UV-Vis spectrum of prepared ZnO NPs and Se/ZnO NPs were obtained within the 200 nm to 400 nm range (Figure 1b). A sharp peak of pristine ZnO NPs was obtained at 361 nm. As a result of doping, a blue shift of the peak was observed at the wavelength of 352 nm, confirming that the doping of the Se metalloid was successfully achieved [25].

### 3.2. XRD Analysis of Pristine ZnO NPs and Se/ZnO NPs

The XRD pattern of pristine and Se/ZnO was obtained, and the size was calculated considering the 2-theta angle of the highest peak (Figure 2). The theta angles obtained were correspondent to the peaks of JCPDS 36-1451 at the 2-theta angles of 31.06°, 33.83°, 35.69°, 47°, 56°, 62.3°, and 67.38° in correspondence with their reflections from the (100), (002), (101), (102), (110), (103), and (200) of wurtzite crystallite of ZnO [25]. Hence, we predict that the peaks obtained are due to the crystalline structure of our ZnO NPs. The concordant peaks of Se/ZnO broadened with respect to the ZnO NPs peaks, depicting that the crystalline nature of ZnO had undergone distortion due to the doping of Se metalloid in ZnO. A lower intensity also showed reduced ZnO wurtzite phase crystallinity due to Se doping. The crystallite size was calculated using the Scherer equation.
(4)D=kλ/(βhkl cos θ )
where k is the constant 0.90. λ is the wavelength of incidence X-ray that is 0.15406 nm. βhkl is the (FWHM) (Table 1) peak width at half maximum and cos θ is the peak position. 

### 3.3. FTIR Analysis of Pristine ZnO NPs and Se/ZnO NPs

The Fourier Transform infrared (FTIR) spectra for two samples, i.e., pristine ZnO NPs and Se/ZnO NPs, was obtained between the range of 550 to 4000 cm^−1^ (Figure 3). This spectrum shows the normal stretching and bending of molecular vibration. The broad peak stretching at the range of 3445 cm^−1^ was shifted to 3421 cm^−1^, and referred to as the molecular vibration of the O-H group, which gives us a clue about Se doping. Another absorption band of the carboxy group C=O group was observed at 1397 cm^−1^ in the case of doped Se/ZnO NPs [25]. The molecular vibrational shift of metal oxide was observed at the lower IR region of the spectra, which is exactly below 600 cm^−1^ [25]. These shifts in the molecular vibrations of the spectra depict that the Se metal is successfully doped in ZnO NPs. The shift in the lower IR region of the spectrum indicated that the bond becomes weak upon doping of Se in ZnO NPs.

### 3.4. SEM Analysis of Pristine ZnO NPs and Se/ZnO NPs

The SEM analysis (Figure 4) of the ZnO NPs and Se/ZnO NPs confirms that the prepared samples are in the nano range and are hexagonal in shape. The SEM of ZnO NPs shows that the particle’s diameter range is between 14.42 nm and 40 nm. In the case of Se/ZnO doped particles, the size was reduced, and the diameter ranged from 14.42 nm to 28 nm. Hence, doping of Se in ZnO NPs resulted in the size reduction of the NPs. 

### 3.5. EDX Analysis of Pristine and Se/ZnO NPs

The strong peaks of Zn and O were shown in the EDS analysis of ZnO NPs (Figure 5a), whereas the significant peaks of Se, Zn, and O were shown in the EDS analysis of Se/ZnO NPs. Figure 5b shows the presence of these elements in the prepared samples. The Zn metal reduction in these NPs indicates that the Se doping is done successfully in the ZnO NPs.

### 3.6. Zeta Analysis

Zeta analysis was done to analyze the charge of prepared NPs. The zeta potential of each NPs of ZnO sample was obtained as +16.1 mV. Similarly, the zeta potential of the NPs of the Se/ZnO sample was obtained as +18.5 mV. NPs are stable because the zeta potential was higher than +15 mV.

### 3.7. Isolation of Bacterial Strains

From the blood agar media, *S. mutans* growth was confirmed by the appearance of darker colonies. The isolation of *E. faecalis* was confirmed by slightly lighter-colored colonies than *S. mutans*. Both species undergo the lysis of erythrocytes on blood agar plates [32]. 

### 3.8. Antibacterial Activity of Pristine ZnO and Se/ZnO NPs

The antibacterial assay of *E. faecalis, S. mutans,* and saliva microcosm showed that the antibacterial activity of 1% ZnO NPs-incorporated discs is higher than the unmodified composite discs. Different concentrations of Se/ZnO NPs were also incorporated in the composite resin discs and tested against the strain of *E. faecalis, S. mutans,* and saliva microcosm (Figure 6a). Results clearly showed that the antibacterial activity was significantly enhanced by the incorporation of Se/ZnO NPs in composite resin against *E. faecalis, S. mutans,* and saliva microcosm (*p* < 0.05) (Figure 6b). 

### 3.9. Biocompatibility Analysis of Pristine ZnO and Se/ZnO NPs

In biocompatibility analysis, the supernatant of the positive control showed 100% hemolysis. In other samples, the supernatant was colorless, showing negligible hemolytic activity towards red blood cells (Figure 7a). 

In quantitative analysis (Figure 7b), the OD of the supernatant was obtained at 545 nm in UV-Vis spectroscopy, and % hemolysis was calculated. It was found that composite materials with Se/ZnO NPs at all concentrations do not have any hemolytic activity as their values lie in the non-hemolytic range, which is less than 5% for materials defined by the ASTM F756 standard. Our positive control (blood with Triton X-100) showed 100% hemolysis, whereas the negative control (PBS solution with blood) had 0% hemolytic activity. The simple composite showed 0% hemolytic activity, and the composite with pristine 1% ZnO NPs showed 5.1% hemolysis that does not fall under the safe range according to ASTM F756, whereas 0%, 1.65%, and 2.32% hemolytic activity was shown by composites with 1%, 2.5%, and 5% Se/ZnO NPs, respectively. 

### 3.10. Mechanical Testing of Pristine and Se/ZnO NPs 

Compressive strength testing revealed that the NP’s incorporation does not affect its mechanical strength. However, there was no significant increase in the mechanical strength between composites with 1% Se/ZnO NPs and 1% of ZnO NPs. It was also revealed that the mechanical strength remains unaffected by increasing the Se/ZnO NPs concentration (Figure 8). 

## 4. Discussion

This study presented the synthesis of pristine and metalloid-doped metal-oxide NPs and their effectiveness as antimicrobial nanofillers in composite resins. The metalloid selected for doping was Se, based on its antioxidant, antiviral, anti-inflammatory, and antibacterial properties [33]. The synthesis of pristine and doped NPs was confirmed by several characterization techniques used with NPs, i.e., UV-Vis spectroscopy, XRD crystallography, FTIR analysis, SEM, and EDX analysis. Zeta potential showed the stability of synthesized NPs.

The UV-Vis spectrograph showed the maximum absorbance at the wavelength of 361 nm in the case of pristine ZnO NPs. A blue shift was observed in the UV spectra of Se/ZnO NPs, which showed the maximum absorption at the wavelength of 352 nm, initially confirming the doping [25,34]. XRD crystallography peaks followed the hexagonal wurtzite structure of ZnO, JCPDS 36-1451, where 2-theta angles of 31.06°, 33.83°, 35.69°, 47°, 56°, 62.3°, and 67.38° were observed and which were in correspondence with their reflections from (100), (002), (101), (102), (110), (103), and (200) peaks. The crystallite size corresponded to the SEM sizes ranging from 14 nm–40 nm for ZnO NPs and 14 nm–28 nm for doped NPs. SEM images also showed no aggregation. Previous studies showed that the smaller the particle size, the easier it is to penetrate the bacterial cell [35]. The decrease in crystallite size can also aid in enhancing antibacterial properties [36]. 

Elemental analysis confirmed the presence of all the desired elements—Zn, Se, and O [25]. Zeta potential confirmed the stability of the prepared NPs as the charges were greater than +15 mV and −15 mV. Due to the high positive charge, the repulsive forces prevented aggregation [37]. Both bacterial species were gram-positive, and the Se/ZnO NPs in our study, which were positively charged, negatively affected their cell wall due to electrostatic interactions [38,39]. These electrostatic forces between the bacterial cell wall and the NPs enhance the antibacterial activity. 

To be an effective nanofiller, the material should not lose the basic properties of pre-existing filling materials. It should exhibit improved antibacterial properties to overcome biofilm formation while retaining its mechanical properties [40]. ZnO NPs have been previously reported as effective antimicrobial agents in dental composites but have limited antibacterial properties at lower concentrations [14,22]. Se was selected as a doping agent due to its ability to enhance the antibacterial properties of ZnO NPs and its biocompatibility with human cells. This study presented the enhanced antibacterial properties of Se/ZnO NPs in dental composite. Enhanced antibacterial activity was also observed in a previous study in which Ag was doped on ZnO and incorporated into resin composites [7,41]. The greater the concentration of NPs in composite resin, the greater the enhancement in the antibacterial activity observed, which was similar to a previous study [7,15]. When tested for cytotoxicity, less cytotoxicity was observed even at higher concentrations of Se/ZnO NPs. Even the highest percentage (5%) showed 2.32% hemolytic activity, which is within the safest range according to the ASTM F756 standard [30]. Limited cytotoxicity of Se/ZnO was also observed in the previous study [42]. Incorporating Se/ZnO NPs did not change the composite resin’s physical properties, hence providing no harm to the aesthetics, which is also important for dental fillings. The mechanical properties of modified composite resin were not affected by increasing the concentration of NPs in composite resin. The load applied on the composite, i.e., 5 kN, was far beyond what a tooth can bear. Hence, mechanical strength was not affected by incorporating Se/ZnO NPs.

The antibacterial activity of composite resin Se/ZnO NPs was highest against the *S. mutans* strain and microcosm at the lowest concentration (1%). They were less cytotoxic and preserved the mechanical strength of the resin composite better than composites with simple ZnO NPs. At 2.5% of Se/ZnO Nps, there was a significant reduction in the growth of *E. faecalis*, at which point both biocompatibility and mechanical strength were preserved. These results showed that composites with Se/ZnO NPs are better antimicrobial restorative materials than the composites containing simple ZnO. 

The enhanced antibacterial activity of Se/ZnO NPs was due to oxidative stress, i.e., by releasing reactive oxygen species (ROS) in the environment, disrupting the bacterial cell wall, and causing the release of cellular organelle in the environment, eventually leading to the breakdown of protein, nucleic acid, and enzymes. The release of metal ions, i.e., Se and Zn ions, may also disrupt bacterial membrane permeability, inhibiting enzyme activity and metabolic activities. [43]

### Limitations

In the present study, there are some limitations to address. The release kinetics and leaching of Zn^2+^ and Se ions were not studied. Furthermore, the thermal stability and NPs distribution within the composite matrix were not studied, which will be explored in future studies.

## 5. Conclusions

This study reported the effectiveness of Se/ZnO NPs in enhancing antibacterial activity and reducing cytotoxicity while maintaining the aesthetics and the mechanical strength of commercial dental composite resin compared to the ZnO NPs incorporated composite resin. Se/ZnO NPs have significantly enhanced antibacterial activity against oral saliva microcosm *E. faecalis* and *S. mutans* microbial strains. They were highly biocompatible even in higher concentrations (5%) and did not negatively impact the mechanical strength of composite resin. Therefore, Se/ZnO NPs are recommended as nanofillers in dental resin composites to prevent composite biofilms. 

## Figures and Tables

**Figure 1 materials-15-07827-f001:**
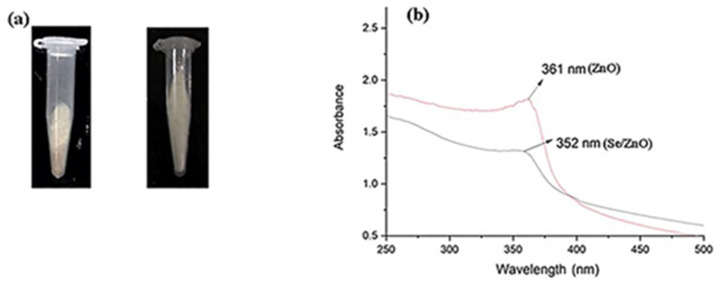
Initial confirmation of ZnO NPs and Se/ZnO NPs synthesis: (**a**) Fine white powdered nanoparticles obtained for ZnO NPs and Se/ZnO NPs and (**b**) UV-Vis spectroscopy of ZnO NPs and Se/ZnO NPs.

**Figure 2 materials-15-07827-f002:**
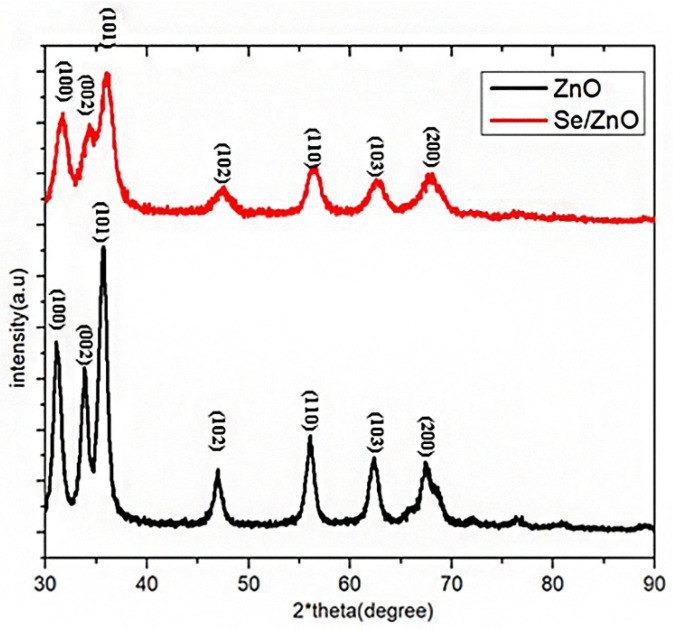
XRD analysis of ZnO NPs and Se/ZnO NPs.

**Figure 3 materials-15-07827-f003:**
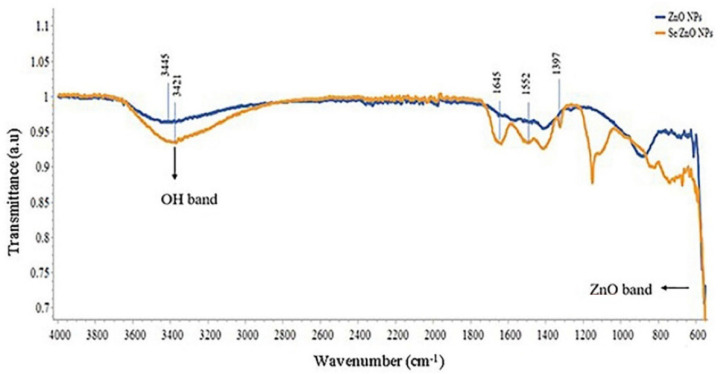
FTIR analysis of ZnO NPs and Se/ZnO NPs.

**Figure 4 materials-15-07827-f004:**
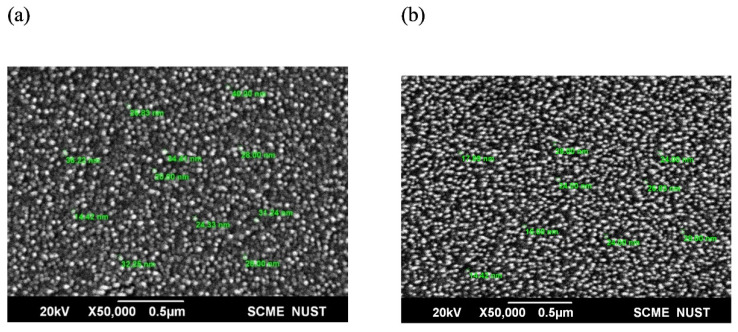
Sem analysis of (**a**) ZnO NPs and (**b**) Se/ZnO NPs at ×50,000 resolution using 0.5 µm scale.

**Figure 5 materials-15-07827-f005:**
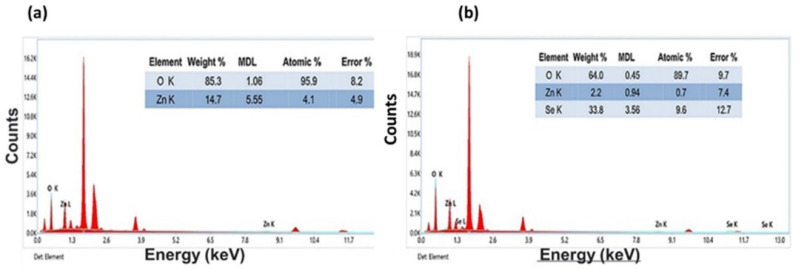
EDX analysis of (**a**) ZnO NPs and (**b**) Se/ZnO NPs.

**Figure 6 materials-15-07827-f006:**
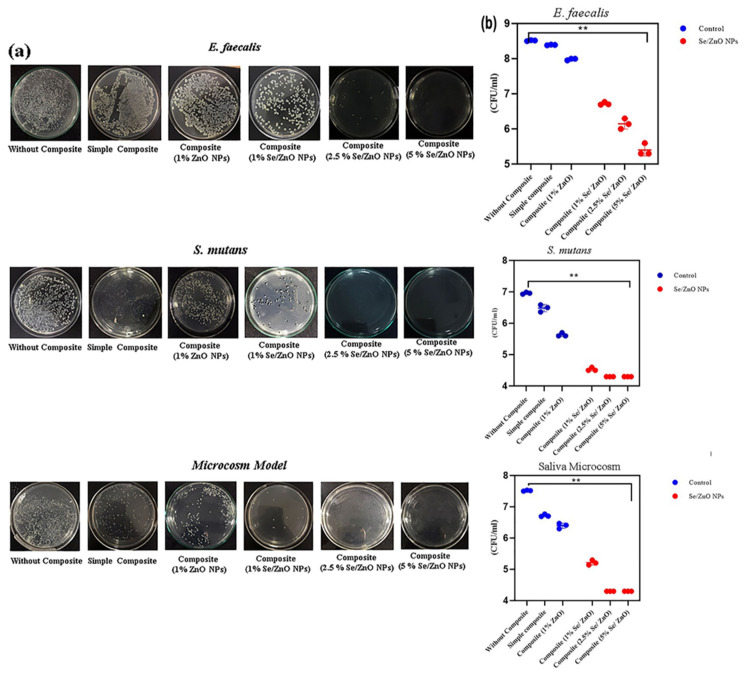
Antibacterial assay of modified composite resin. (**a**) Antibacterial activity of modified composite resin discs on TSA plates and (**b**) CFU/mL of *E. faecalis, S. mutans,* and Saliva microcosm models. (** = *p* < 0.01).

**Figure 7 materials-15-07827-f007:**
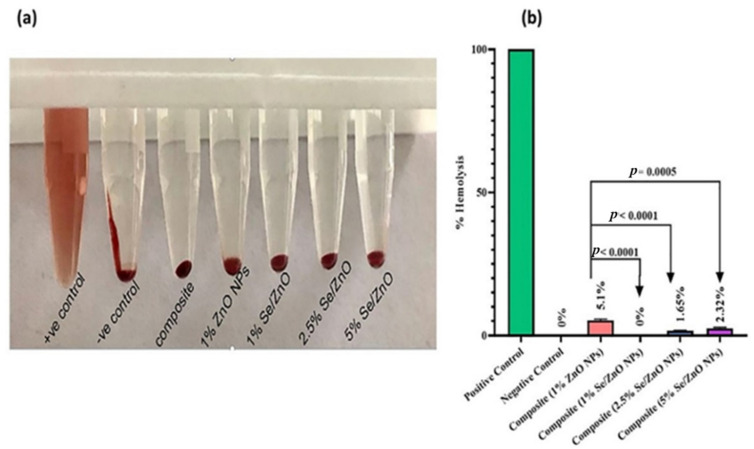
Hemolytic activity of resin composite ZnO NPs and Se/ZnO NPs. (**a**) Supernatant does not show any hemolysis of RBCs except positive control that shows 100% hemolysis of RBCs and (**b**) statistically significant reduction in hemolysis (*p* < 0.005).

**Figure 8 materials-15-07827-f008:**
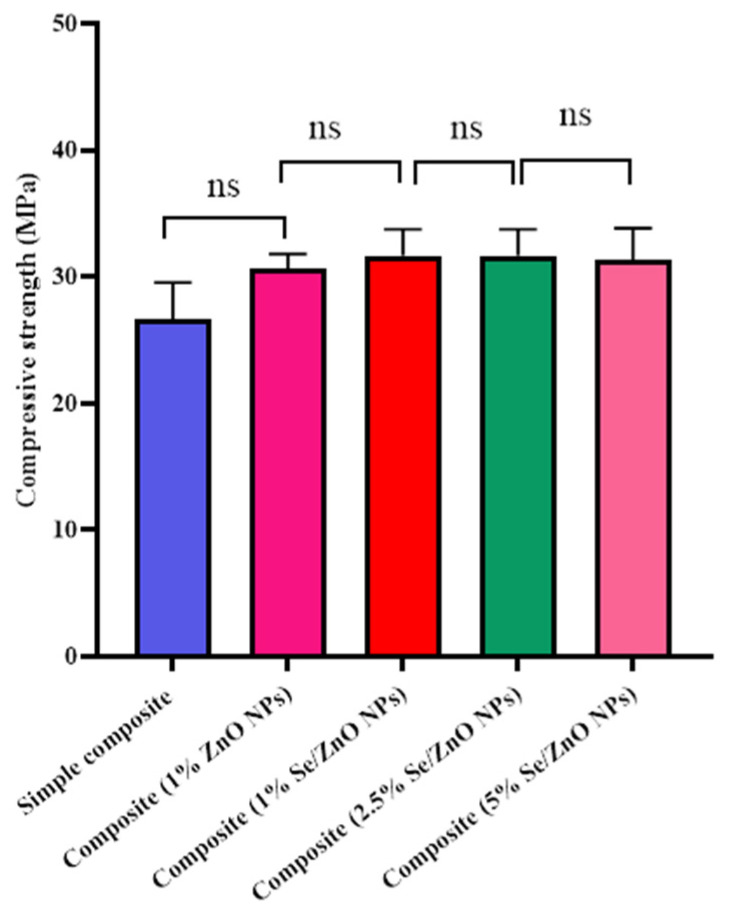
Mechanical strength not affecting by increasing the concentration. ns = not significant.

**Table 1 materials-15-07827-t001:** Crystallite size of pristine ZnO and Se/ZnO.

Sr. No	Nanoparticle	2θ°	β _(FWHM)_	D(nm)	Size (nm)
1.	Pristine ZnO	35.63°	0.19	2.51	42.35
2.	Se/ZnO	36.50°	0.29	2.54	28.64

## Data Availability

Not applicable.

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
