# Peer review of "Effectiveness of Se/ZnO NPs in Enhancing the Antibacterial Activity of Resin-Based Dental Composites"

_materials, 2022, doi:10.3390/ma15217827_

Round 1
Reviewer 1 Report
The authors should checked and revised manuscript.
Composite materials should be defined in details.
Effect of nanoparticles in the composite resin should be explained ( effect of mechanical stability, thermal behaviour )
Interaction between NPs and composite resin matrix should be explained
Phase separation and leaching out effect of NPs should be determined
Reviewer 2 Report
The manuscript describes the investigation of the effectiveness of Se doped ZnO (Se/ZnO) NPs as an antibacterial nanofiller and its impact on mechanical properties of the resin composites.
The novelty is not apparent, as ZnO NPs incorporated dental resins have been previously investigated and the mechanochemical synthesis of Se/ZnO NPs has also been published as well as its antibacterial properties. Please specify the novelty of this work.
The scope is narrow as only one type of dental restorative material, resin, was investigated, on the influence of incorporation of NPs, suitable for a dentistry journal; although the methods are appropriate for achieving the objectives set out, some key information, such as operating parameters applied for FTIR, XRD and SEM are missing; the result dissemination is appropriate but only within the narrow scope, thus the overall impact is low.
English usage needs to be improved significantly as there are many grammatical mistakes and problematic usage, to name a few “its impact of mechanical properties” “Mechanical strength not affecting by increasing the concentration” “are not seemed to be affected”” values lies”
Hence, a major revision is necessary.
Some suggestion:
1. To suit publication in a general materials science journal like Materials, it is suggested to expand the scope by introducing resin in the context of the general literature of dental restorative materials, as such, for example:
L41, Besides the biocompatible and anticariogenic glass ionomer [ref], composite resin has gained advancement and popularity in recent years due to its physical and mechanochemical properties….
[ref] Orsolya Lang, Laszlo Kohidai, Zsofia Kohidai, Csaba Dobo-Nagy, Krisztian B. Csomo, Mira Lajko, Miklos Mozes, Sandor Keki, Gyorgy Deak, Kun V. Tian, Veronika Gresz, Cell physiological effects of glass ionomer cements on fibroblast cells, Toxicology in Vitro, 61, 2019, 104627.
2. “estimated annual cost of restorative materials in 2005” is outdated, please use the latest statistics.
3. L100, confirmation can not be done “by” a technique, but can be done with it, by a person.
4. There are many cariogenic bacteria, why did you choose S. mutans and E. faecalis?
5. A space is needed between numbers and units.
6. Equations must be numbered.
7. L161 What is CS?
8. L174, typo “posiitive control.”
9. Fig. 3, 5 resolution must be increased. Fig. 4 caption lacks magnification applied and scale bar length.
10, How was the biocompatibility accessed, rather than applying the common cytotoxicity tests using cell lines like was done in [ref]?
Reviewer 3 Report
Manuscript materials-1900161 is well written and deserves publication after minor revision.
Some suggestion to improve the manuscript:
1. Emphasize novelty at the end of the abstract and of originality at the end of the introduction.
2. Section 3.1: compare your data with literature information.
3. What is the nature of the C=O group vibration at 1397 cm-1 (line 225). Give reference information.
4. Increase text resolution of Fig. 1b, 3, 4 (green insertions are illegible), 6 and 7b. Use a unitary style for figures format. Curves of Fig 1b and axis of Fig 3 and 5 cannot be seen.
5. Give units for all the data obtained in Table 1.
6. Correct typos:
- Line 125: 400 mV/cm2 should be 400 mW/cm2
- Line 224: se doping should be Se doping
- Line 227: rage should be range
Author Response
Dear Reviewer #3, Many thanks for your valuables comments and suggestions which have contributed to improve the manuscript further. We have now addressed each of your queries in the attached point-by-point rebuttal letter. All corrections have been applied to the manuscript which has been also attached to this mail. Hope that the manuscript fits better your expectations and will be considered for publication. We thank you very much for your time, consideration, and expertise.
Round 2
Reviewer 2 Report
The overall quality has been improved after the revision.
The result presentation needs to be improved further.
All the figures must achive the same clarity as Fig. 8.
Fig. 1b, too much blank space, reduce y-axis range to 2.5, remove the label box and simply label the curves.
Fig. 3, check x-axis label, -1 should be superscripted.
fig. 4, can not read labels.
fig. 5, hard to see.
fig. 6a, hard to see. Move the same repetative labels, simplely use them once and above each vertical column. Enlarge each image. Reduce b size, better to align the 3 vertically so for each bactetia, images and CFU result are horizontally aligned.
Fig. 8, strength unit is Pa or MPa? It appears overlow at the present values.
M&M does not have a statistical analysis section. Please add it.
Equation numbers should be right-aligned.
Author Response
Please see attachement
